



# Hyperspectral reflectance dataset of dry, wet and submerged marine litter

Els Knaeps[1], Sindy Sterckx[1], Gert Strackx[1], Johan Mijnendonckx[1], Mehrdad Moshtaghi[1], Shungudzemwoyo P. Garaba[2], Dieter Meire[3]

[1] Flemish Institute for Technological Research (VITO), Belgium
[2] Marine Sensor Systems Group, Institute for Chemistry and Biology of the Marine Environment, Carl von Ossietzky University of Oldenburg, Schleusenstraße 1, Wilhelmshaven 26382, Germany
[3] Flanders Hydraulics Research, Berchemlei 115, 2140 Antwerp, Belgium

*Correspondence to*: Els Knaeps (els.knaeps@vito.be)

**Abstract**

This paper describes a dataset consisting of 47 hyperspectral reflectance measurements of plastic litter samples. The plastic litter samples include virgin and real samples from the Port of Antwerp. They were measured in dry conditions in the VITO calibration facility and a selection of the samples was also measured in wet conditions and submerged in a watertank at Flanders Hydraulics. The construction on top of the tank allowed to submerge the plastics and keep sediments in suspension. The

spectral measurements were performed using an Analytical Spectral Devices (ASD) FieldSpec 4 and a Spectral Evolution (SEV) spectrometer. The datasets are available on the 4TU.ResearchData open-access repository (ASD dataset: https://doi.org/10.4121/12896312.v2, Knaeps et al., 2020; SEV dataset: https://doi.org/10.4121/uuid:9ee3be54-9132-415a-aaf2-c7fbf32d2199, Garaba et al., 2020).

## 1 Introduction

Spectral reflectance measurements of plastic specimen have been collected in the framework of the Hyperspectral remote sensing of marine plastics (Hyper) project. The data was used to provide guidance on the selection of appropriate wavelengths for marine plastics identification. Spectral reflectance of dry plastic polymers is known and this knowledge is used in the industry for sorting plastics from other litter and discriminate between different polymers (Huth-Fehre et al., 1995;Masoumi et al., 2012). Knowledge on spectral reflectance of marine litter, which is wet or submerged, is however limited. To this end,

there has been a rising interest in establishing spectral reference libraries of plastic litter in different conditions and state. More recently, a dataset of the spectral reflectance of marine harvested and virgin plastics which were measured in dry and wet conditions was reported and discussed in open-access (Garaba and Dierssen, 2020). The Hyper dataset is an extension to the existing datasets because it includes spectral measurements of additional plastic specimen and involves simulated scenarios of submerged samples with varied water clarity. The dataset has the potential to help to develop appropriate algorithms for marine

plastic detection, identification and select appropriate sensors relevant for marine plastics.


In total, 47 plastic specimen were measured with the Analytical Spectral Devices (ASD) FieldSpec 4 spectrometer. Six of them were also submerged in a controlled way in a water tank. The plastic specimen consist of virgin plastics and real samples from the Port of Antwerp. The dataset has been made publicly available (Knaeps et al., 2020: https://doi.org/10.4121/12896312.v2). For comparison, a few plastic specimen have also been measured with a Spectral Evolution (SEV) SR-3501 hyperspectral radiometer with its own preprocessing applied. The SEV dataset is also publicly available (Garaba et al., 2020: https://doi.org/10.4121/uuid:9ee3be54-9132-415a-aaf2-c7fbf32d2199).

## 2 Laboratory and tank set-up

Measurements of the spectral reflectance were performed at two locations. The first was an optical calibration laboratory at VITO, Belgium. The spectral observations of dry plastics were conducted in the dark laboratory room equipped with 2 halogen tungsten lamps. Further experiments were done using a water tank at Flanders Hydraulics facility in Antwerp, Belgium (**Figure 1**). The water tank (diameter = 2 m, depth = 3 m) was customized to simulate and assess the optical properties of submerged plastics at fixed water depths and changing water clarity. The tank was equipped with a propeller that allowed mixing of the water to create near homogenous distribution of sediments. A set of two halogen tungsten lamps (12V 50W GY9.5 - Original Gilway L9389) were used to provide artificial lighting simulating sunlight.

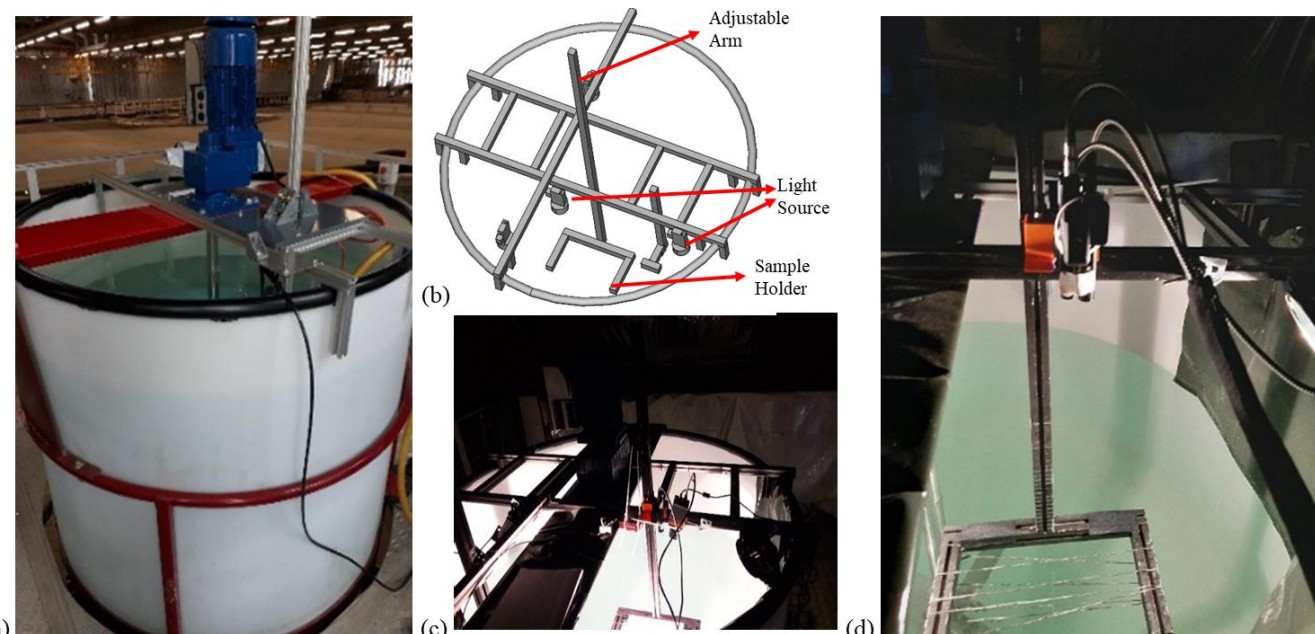

**Figure 1. (a) Original set-up of the tank, (b) schematic of the aluminium frame, (c) experimental set-up of the tank inside covered with black cloth, and (d) foreoptic ends of the ASD and SEV spectroradiometers.**





A tailor-made aluminum frame was attached to the water tank for the attachment of the spectroradiometer detector, lights and samples (**Figure 1b**). The frame was also painted black so that it would not contribute to the bulk spectral signal expected from only the plastic targets. An adjustable arm was integrated on the frame with a holder for the targets and predetermined depth markings (**Figure 1c**). A black cloth was used to create a dark surrounding over the water tank to mitigate unwanted

straylight from the laboratory surfaces. Furthermore, the side walls of the tank were covered with a black plastic to mitigate background reflections, no background correction was applied to the measurements as it was determined to be negligible. The foreoptic end of the ASD and SEV radiometers were attached to the aluminium frame on top of the tank to make sure the viewed the same target at the same distance (**Figure 1d**).

## 3 Data collection

Data collection consisted of spectral reflectance measurements of different plastic specimen, both virgin plastics and plastic waste from the Port of Antwerp. Spectral measurements were performed by two different teams using different instruments (ASD FieldSpec 4 and SEV SR-3501 hyperspectral radiometer). In the tank, sediments were added and the Total Suspended Matter (TSM) concentration was analyzed after each set of measurements.

### 3.1 Spectral measurements

Spectral reflectance of the plastic targets was measured with an ASD FieldSpec 4 and a SEV SR-3501 hyperspectral spectroradiometer from the ultraviolet (UV) to shortwave infrared (SWIR). The ASD spectrometer has a spectral resolution of approximately 3 nm at around 700 nm. The spectral resolution in the SWIR varies between 10 nm and 12 nm. The SEV spectroradiometer has spectral resolution between 4 and 10 nm. For both instruments, we derived a relative hemispherical directional reflectance ($R$) of the sample which was normalized to a 99 % Labsphere Spectralon® Lambertian panel. In the

tank, the spectralon panel was placed at the same position as the plastic specimen was placed, i.e. on the holder of the adjustable arm. For brevity, additional specifications are summarized below (**Table 1**).

**Table 1: Hyperspectral radiometer specifications and settings during the study.**

|  | ASD | SEV |
|---|---|---|
| Spectral range (nm) | 350-2500 | 280-2500 |
| Spectral resolution (nm) | VNIR: ca. 3 nm<br>SWIR: 10-12 nm | 4 nm (280–1000 nm)<br><= 10 nm (1000–1900 nm)<br><= 7 nm (2100–2500 nm) |
| Scans per measurement | 30 | 30 |
| Replicate measurements | 5 | 5 |
| Foreoptic field-of-view | 8 ° | 8 ° |



Processing of the ASD data included calculation of the mean and standard deviation from the replicate measurements. No additional smoothing was applied on the data. The SEV measurements were smoothed using a third order polynomial Savitzky-Golay least-square algorithm at frame length of 31. No splice correction was applied to the data to compensate for typical steps in spectra due the transition between the three detectors. We computed the Unbiased Percentage Difference (UPD) as a measure of uncertainty between the two spectroradiometers

$$UPD(\%) = 200 * \left( \frac{\left| R_\lambda^{ASD} - R_\lambda^{\overline{SEV}} \right|}{R_\lambda^{ASD} + R_\lambda^{SEV}} \right) \qquad (1)$$

Where $R$ is the reflectance at wavelength ($\lambda$) the ASD and SEV respectively.

### 3.2 TSM measurements

Water clarity was changed by adding fine clay collected from Deurganckdok, a tidal dock in the harbor of Antwerp, connected to the Scheldt river (B). The clay had a median particle size $D_{50}$ of $11.0 \pm 0.3$ µm and ranged between a $D_{10}$ of 2 µm to a $D_{90}$ of 51 µm. Initial water clarity was high meaning no suspended material, it was then followed by a adding a small amount clay to achieve medium and finally low visibility with a high concentration of TSM. The sediment concentration was measured by taking water samples of 1 L at the water surface, and filtration of the samples.

### 3.3 Plastic specimen

Optical properties were measured on a set of real/weathered and virgin plastics. Weathered marine litter was gathered from the port of Antwerp consisting of plastic bottles, plastics bags, rope, drinking containers and wood (**Figure 2**). Virgin plastics (off the shelf) consisted of ropes, bags, placemats, bottles, cups and foam. We gathered blue, yellow, pink and orange placemats as well as orange, blue and white ropes. The ropes were prepared for the reflectance measurements in three ways: compact on a roll as bought in the store, unrolled and nicely aligned along a frame (**Table 2**).

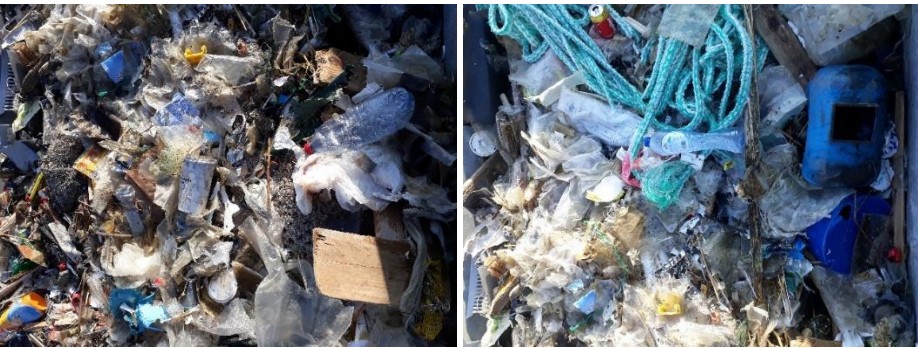

**Figure 2. Litter specimen found in the Port of Antwerp, Belgium.**





The following naming convention (***name_state_depth_TSM***) was utilized to identify the type or ***name*** of plastic specimen, ***state*** of sample as ***d***(dry), ***w***(wet) or ***s***(submerged), the ***depth*** below water surface [cm] and ***TSM*** concentration [mg L$^{-1}$]. For example, OrangePlacemat_s_4_200 refers to measurements of an orange placemat submerged at 4 cm depth and a TSM of 200 mg L$^{-1}$. There are 4 different polymer types within the dataset: polyethylene terephthalate (PET), Low-density polyethylen (LD-PE), Polyester(PEST) and polypropylene (PP). A total of 47 materials were investigated and the descriptors as well as metadata is summarized below (**Table 2**).

**Table 2: Overview of measurements**

| Type | Plastic specimen | | Polymer type | Available measurements (d=dry, w=wet, s=submerged) |
|---|---|---|---|---|
| Bottles | | | | |
| virgin | bottle filled with water 1 | | PET | d |
| virgin | bottle filled with water 2 | | PET | d |
| virgin | bottle empty and crushed 1 | | PET | d |
| virgin | bottle empty and crushed 2 | | PET | d |
| virgin | bottle empty and crushed 3 | | PET | d |
| virgin | bottle empty and crushed 4 | | PET | d |
| virgin | Bottles on frame | | PET | w |
| PET Cups | | | | |
| virgin | White PE-LD cup flat | | LD-PE | d |
| virgin | White PE-LD cup straight | | LD-PE | d |





| Placemats | | | | |
|---|---|---|---|---|
| virgin | Orange placemat | | Not specified | d, w, s |
| virgin | Blue placemat | | Not specified | d, w, s |
| virgin | Pink placemat | | Not specified | d, w, s |
| virgin | Yellow placemat | | Not specified | d, w, s |
| Ropes | | | | |
| virgin | Orange PP rope on a roll | | PP | d |
| virgin | Orange PP rope unrolled | | PP | d |
| virgin | Orange PP rope aligned around frame | | PP | d, w, s |
| virgin | Blue PP rope on a roll | | PP | d |
| virgin | Blue PP rope unrolled | | PP | d |
| virgin | Blue PP rope aligned around frame | | PP | d, w, s |
| virgin | White PP rope on a roll | | PP | d |
| virgin | White PP rope unrolled | | PP | d |
| virgin | White PP rope aligned around frame | | PP | d, w, s |
| virgin | White polyester rope on a roll | | Polyester | d |



| virgin | White polyester rope unrolled | | Polyester | d |
|---|---|---|---|---|
| **other** | | | | |
| virgin | garden net | | Not specified | d |
| virgin | Green foam | | Not specified | d |
| **bags** | | | | |
| virgin | white transparant plastic bag 1 | | Not specified | d |
| virgin | white transparant plastic bag 1 wrinkeled | | Not specified | d |
| virgin | white transparant plastic bag 2 | | Not specified | d |
| virgin | white transparant plastic bag 2 wrinkeled | | Not specified | d |
| virgin | Black plastic bag | | Not specified | d |
| virgin | Black plastic bag on frame | | Not specified | w |
| virgin | white transparant plastic bag 1 on frame | | Not specified | w |
| **Port of Antwerp** | | | | |
| real | grey cloth | | Not specified | d |
| real | Waste rope | | Not specified | d |
| real | waste blue plastic bag | | Not specified | d |
| real | waste green rope | | Not specified | d |



| real | orange tube | | Not specified | d |
|------|-------------|--|---------------|---|
| real | transparant wrapping foil | | Not specified | d |
| real | Pellets | | Not specified | d |
| real | thin isolating wrapping foil | | Not specified | d |
| real | Extended Polystrene(EPS) | | Not specified | d |
| real | Energy drink container | | Not specified | d |
| real | Wood 1 | | Not specified | d |
| real | Wood 2 | | N/A | d |
| real | Wood 3 | | N/A | d |
| real | Wood4 | | N/A | d |

### 3.4 Tank scenarios

The tank experiment involved carefully submerging a selection of plastic samples to fixed depths. The selected samples include the new placemats (orange, blue, pink, yellow) and PP ropes (orange, blue, white ) bound to a plexiglass frame making a uniform flat target. First tests were performed with clear water before adding the clay sediments. The plastics were measured first in dry conditions in the tank just above the water surface followed by submerging to fixed depths of 2.5 as just below water surface, 5, 9, 12, 16 and deepest at 32 cm. Finally, the plastics were measured again just above the water surface as wetted plastics. These steps were repeated after adding sediments for the two scenarios of moderate and high TSM concentrations.





## 4. Results and discussion

### 4. 1 TSM results

Laboratory analyses of the water samples taken from the tank after dilution 1 showed a TSM concentration of 75 mg/L and a TSM concentration of 321 mg/L after dilution 2 (**Table 2**).

**Table 3: TSM concentrations**

| 12/12/2019 | Time UTC | Sample 1 [mg L$^{-1}$] | Sample 2 [mg L$^{-1}$] | Sample 3 [mg L$^{-1}$] | Mean ± st.dev [mg L$^{-1}$] |
|---|---|---|---|---|---|
| Dilution 1 | 12:53:00 | 79 | 74 | 72 | 75 ± 3.6 |
| Dilution 2 | 14:02:00 | 315 | 325 | 324 | 321 ± 5.5 |

### 4.2 Reflectance results

#### 4.2.1 Tank water

The water in the tank was measured with the ASD spectrometer without any plastics in the water. 'Clear' tank water was measured on two dates: 18 October 2019 and 12 December 2019. Both water spectra are shown in Figure 4 in black and dashed black line. In the visible region, the tank water measured on 18 October has a higher reflectance than the tank water measured on 12 December. In the NIR and SWIR, the water from 12 December has a clear reflectance signal which is not expected because of strong pure water absorption. The reason for the difference in reflectance between both dates can be some remaining

sediments in the water and some dust floating on the surface on 12 December, meaning that the water was not completely clear. On 18 October the water was clearer and the white bottom of the tank was visible, which can contribute to the reflectance signal. On 12 December the white bottom was less visible and the dust on the surface might cause the non zero reflectance in the NIR and SWIR. It is possible to account for this remaining reflectance in the NIR and SWIR by subtracting the reflectance at 1200 nm as is also done to account for residual sky glint reflection in outdoor water reflectance measurements (Knaeps et

al., 2015). The result is shown in Figure 4 by the black dotted line. The reflectance of the turbid tank water with TSM concentrations corresponding to 75 and 321 mg L-1 is also shown in (**Figure 3**). Increase in TSM concentration results in an overall increase in the spectral reflectance. The NIR is no longer zero and a local maximum reflectance is observed around 1070 nm in the SWIR.





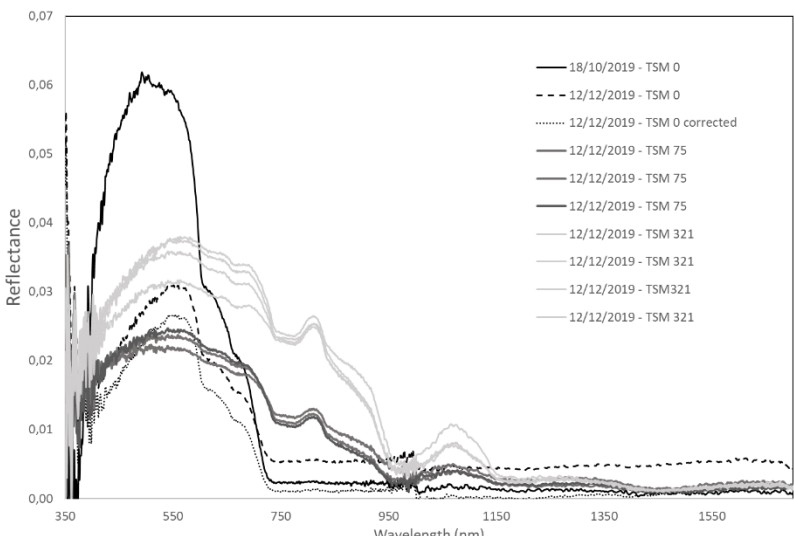

**Figure 3. Spectral reflectance of tank water with various TSM.**

### 4.2.2 Placemats

5  The reflectance in dry conditions from the VIS to the SWIR can be seen in (**Figure 4**). In the VNIR spectral range, all placemats look very different and the inherent spectral reflectance shape in the visible spectrum was consistent with the apparent colour of the samples. Overall, the white placemat had the highest reflectance and in the SWIR, the spectral shapes of the specimen were identical. Furthermore, the absorption features in the SWIR spectrum were located at the same wavebands suggesting these placemats shared similar polymeric composition. The strongest absorption features were observed around 1216 nm, 1397

10  nm and 1730 nm. Absolute reflectance differs considerably, in particular between 1000 and 1700 nm.

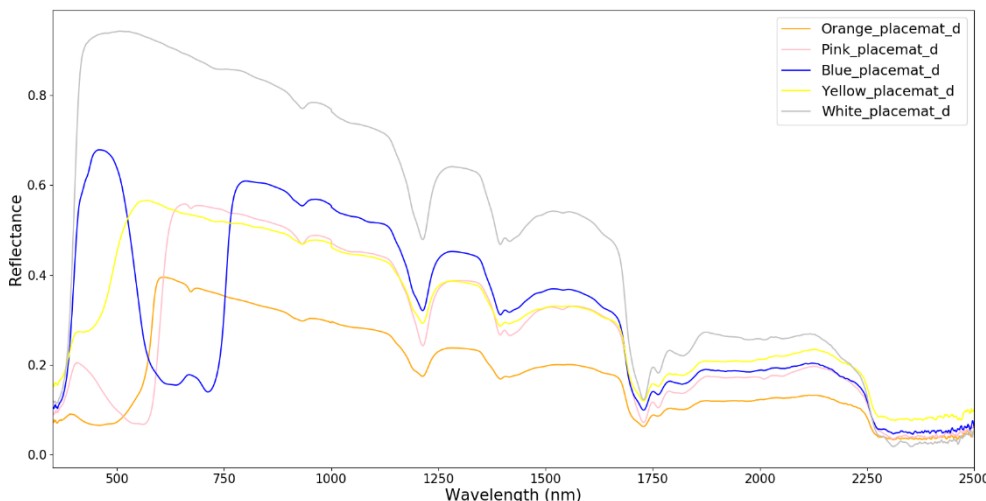

**Figure 4. Spectral reflectance of dry plastic placemats.**

The mean and standard deviation, as shown in Figure 5, was calculated based on 5 replicate measurements. The dry
5  measurements were collected in the VITO calibration facility and the replicate measurements were obtained by slightly
changing the position of the plastic target. Hence, very homogeneous targets will have a smaller standard deviation. The mean
and standard deviation for the blue and orange placemat was very small: 0.00056 and 0.00283 for the blue placemat at 400
and 900 nm respectively 0.00484 and 0.00690 for the orange placemat at 400 and 900 nm respectively (**Figure 5**). The small
standard deviation was also used as a prerequisite for submersion in the tank because measurements at the different depths
10  should not be influenced by differences in the spectral reflectance of the plastic specimen itself (due to e.g. inhomogeneities
of the surface).



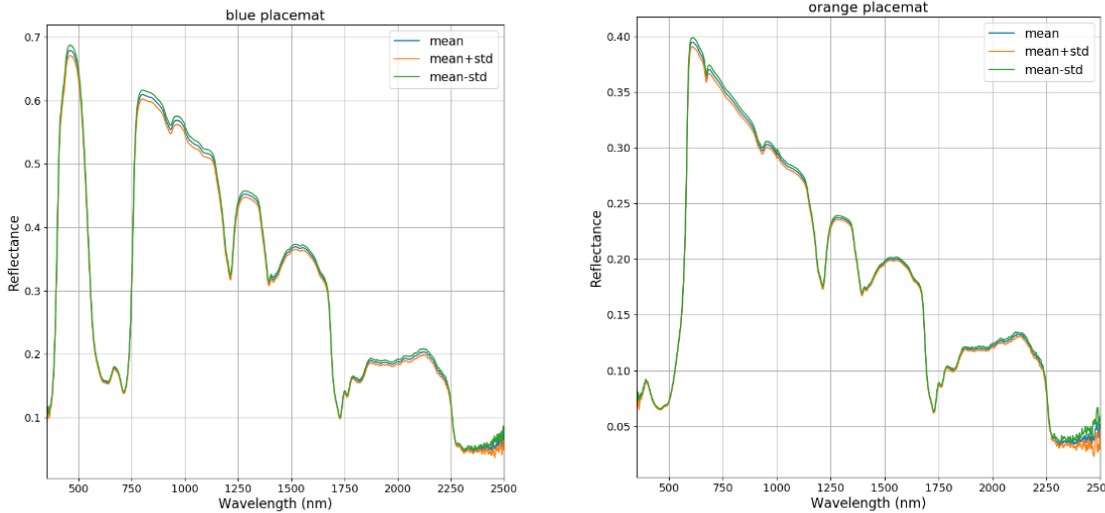

**Figure 5. Mean and standard deviation for the dry blue and orange placemat.**

The spectral measurements of the orange placemat when submerged in the water tank, in clear conditions and with different
dilutions of sediment (**Figure 6**). In addition, the first figure includes a dry measurement in blue and a wet measurement in
grey. The figures show the decrease in reflectance when the plastic is wet and submerged and the changes to the shape of the
spectrum, particularly in the NIR. Between 900 and 1000 nm, the absorption of water strongly increases causing a rapid
decrease in reflectance detected from the submerged targets both at deep (32 cm) and in shallow (2.5 cm) water. Compared to
the neighbouring wavebands between 1000 and 1140 nm, absorption by water is lower resulting in reflectance peaks of the
plastic targets over this spectrum range. Table 4 shows, for all the depths and for different turbidity, at which wavelength the
reflectance reaches a threshold of 0,001. Beyond this wavelength, reflectance will always be lower than 0,001, and the plastic
target cannot be detected. For instance, when the orange placemat is submerged in clear tank water at a depth of 8 cm,  there
will be no measurable reflectance beyond 1134 nm. These values are specific for the orange placemat and will be different for
other plastic specimen. The reader can easily produce similar values based on the published dataset.  The reflectance threshold
was subjectively chosen and the reader is also free to use a different one. For the clear tank water, measurements from
18/10/2019 and 12/12/2019 are included. Although TSM concentrations are not exactly known, it has been shown (Figure 4)
that on 18/10/2019 the water in the tank was clearer and there was remaining dust and/or sediment in the tank on 12/12/2019.

The values in Table 4 for the clearest tank water (18/10/2019) show that SWIR wavelengths between 1000 and 1415 nm
provide information on the orange placemat when slightly submerged (1 up till 18 cm). These results should however be
confirmed in an outdoor setting under different lightning conditions and presence of surface features. With sediment in





suspension, the reflectance of the orange placemat will be slightly or fully masked by the reflectance of the sediments. As shown in the table, there is measurable reflectance in the SWIR, even with the orange placemat at 32 cm depth. Comparing these values with the values of the clear tank water shows that the reflectance at these longest wavelengths is coming from the sediments in the water.

**Table 4: Wavelength corresponding to a reflectance threshold of 0,001 for the orange placemat.**

| Wavelength (nm) | Depth (cm below the water surface) | | | | | | | | | | | | |
|---|---|---|---|---|---|---|---|---|---|---|---|---|---|
| | 1 | 2 | 2.5 | 3 | 4 | 5 | 8 | 9 | 12 | 13 | 16 | 18 | 32 |
| **Clear tank water** | 1415 | 1324 | | 1287 | 1162 | | 1134 | | | 1090 | | 1058 | |
| **Clear tank water** | | | 1153 | | | 1147 | | 1124 | 1098 | | 1079 | | 1051 |
| **75 mg L-1** | | | 1151 | | | 1151 | | 1140 | 1142 | | 1138 | | 1156 |
| **321 mg L-1** | | | 1358 | | | 1315 | | 1322 | 1314 | | 1303 | | 1308 |

a)

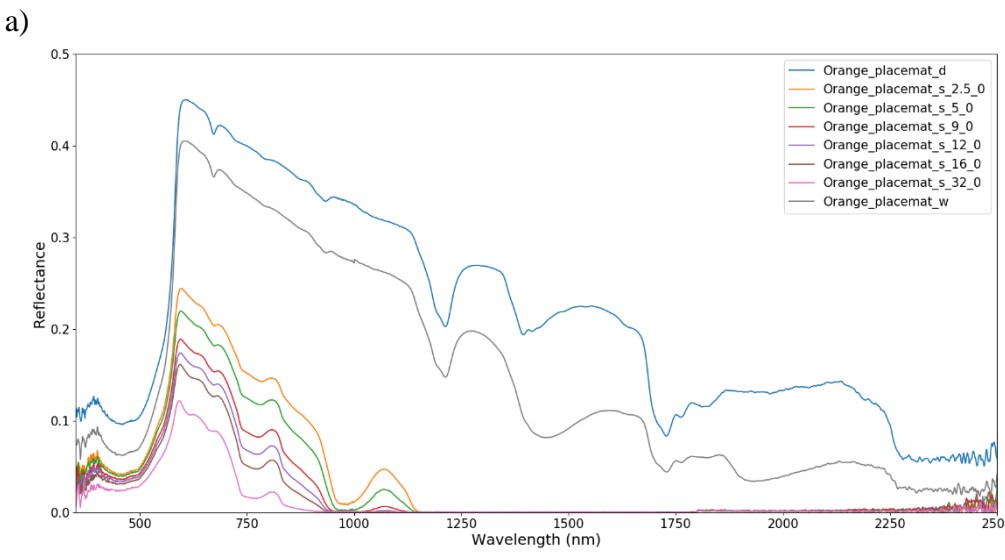

10  b)



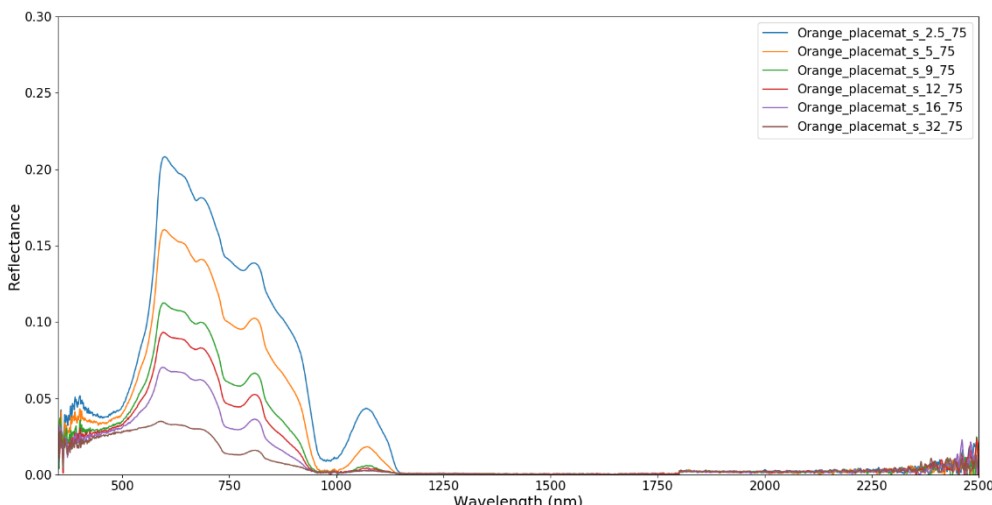

c)

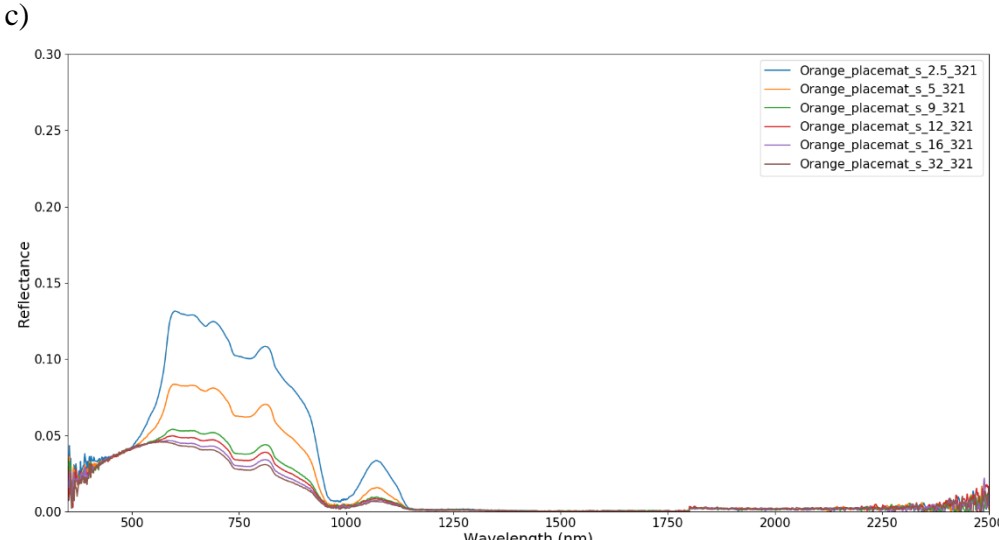

**Figure 6. Spectral reflectance of the orange placemat submerged in a) clear water, b) water with a TSM concentration of 75 mg/L and c) water with a TSM concentration of 321 mg/L.**

### 4.2.3 Ropes

The white polyester rope (unrolled) was the most reflective item in this group, particularly in the visible. The blue and orange rope had a lower reflectance in the visible with spectral shapes in agreement with the apparent colour (**Figure 7**). Three of the ropes are polypropylene polymers, which can also be observed from the spectral absorption features in the SWIR. The white polyester rope has different features in the SWIR. Additional measurements were conducted on the off the shelf/rolled, unrolled ropes and the rope aligned on the plexiglass. It was noted that the rolled rope had the highest reflectance (**Figure 7b**). One of the rolled ropes even had a reflectance above 1, probably due to the exact position of the ASD fibers pointing towards a highly



reflective glint area on the rope. The standard deviation is smallest for the rope aligned along the frame (**Figure 8**). The rope aligned on a plexiglass frame was also used for submersion in the tank. In some of the spectra, a small jump can be observed around 1000 nm, and a second smaller one around 1750 nm. This is caused by the design of the ASD, consisting of three spectrometers and a fiber optic which is actually a bundle of 57 individual fibers, randomly oriented (Analytical Spectral

5   Devices, 1999). VNIR and SWIR fibers are organized differently which makes that different areas of the surface are observed with different parts of the spectrum" resulting in steps in spectra at the joins between each detector (Arthur et al., 2012). This effect is observed when targets are non-uniform. It was decided not to apply any smoothing on the spectra because the jumps can provide information on the uniformity of the target.

10   a)

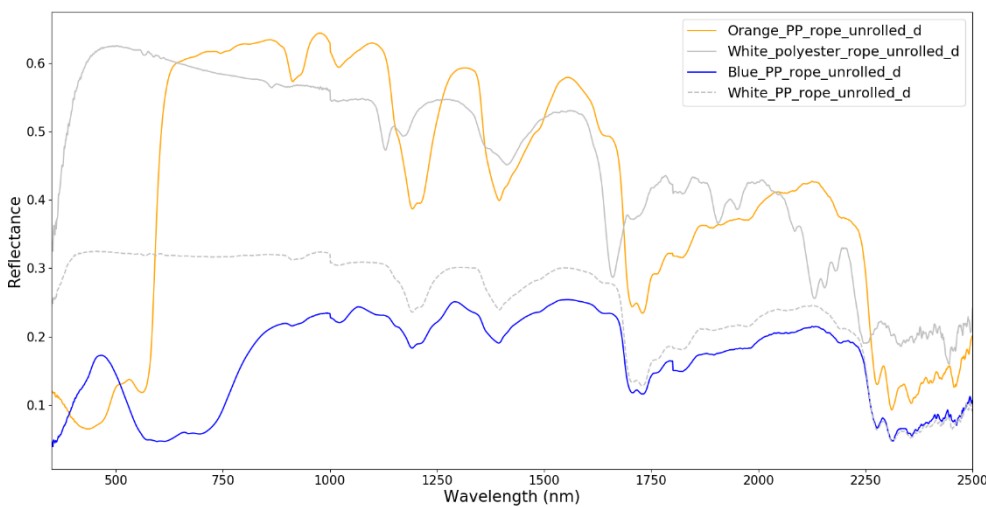

b)



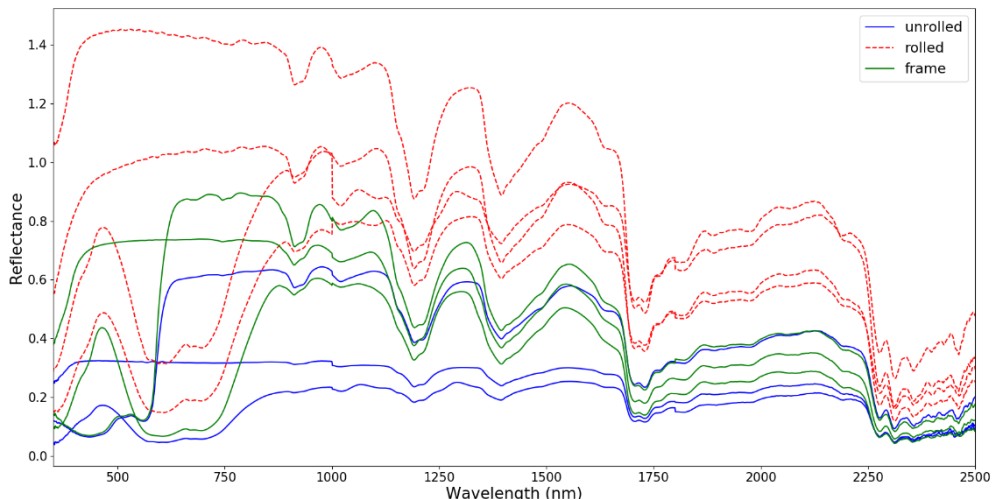

**Figure 7. Spectral reflectance of dry ropes, a) white, orange and blue rope, b) all ropes in different conditions: on a roll as bought in the store, unrolled and aligned on a plexiglass frame.**

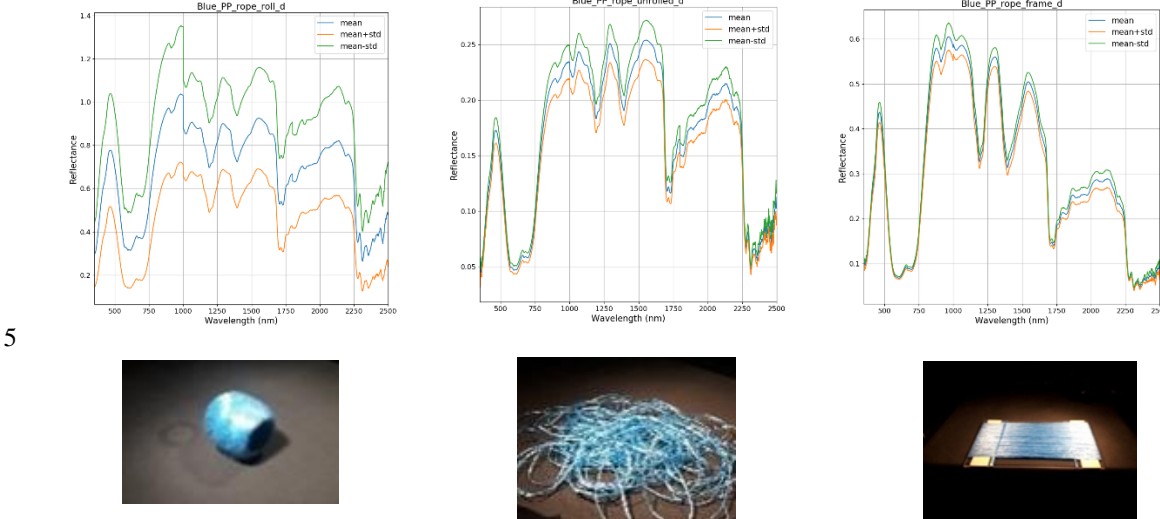

10 **Figure 8. Mean and standard deviation for the blue rope in 3 conditions: on a roll as bought in the store, unrolled and aligned on a plexiglass frame.**





a)

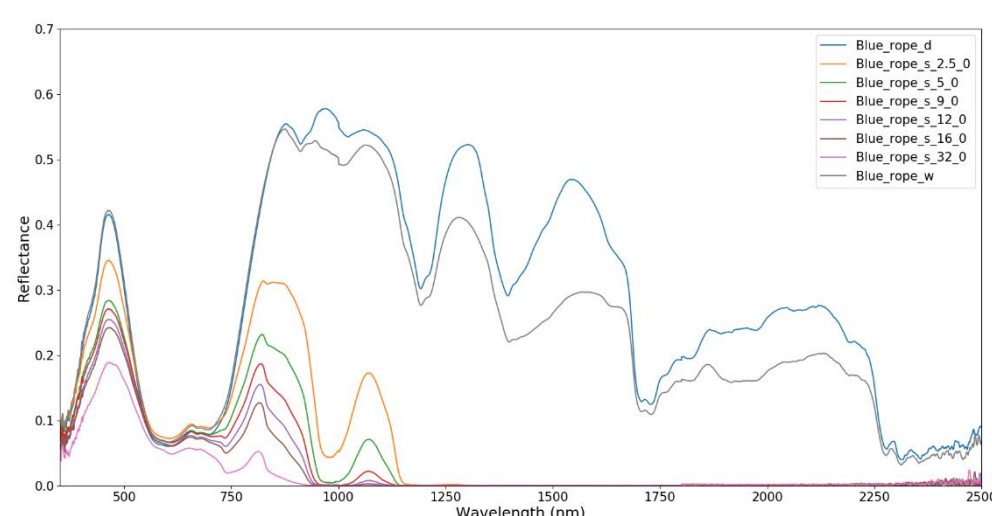

b)

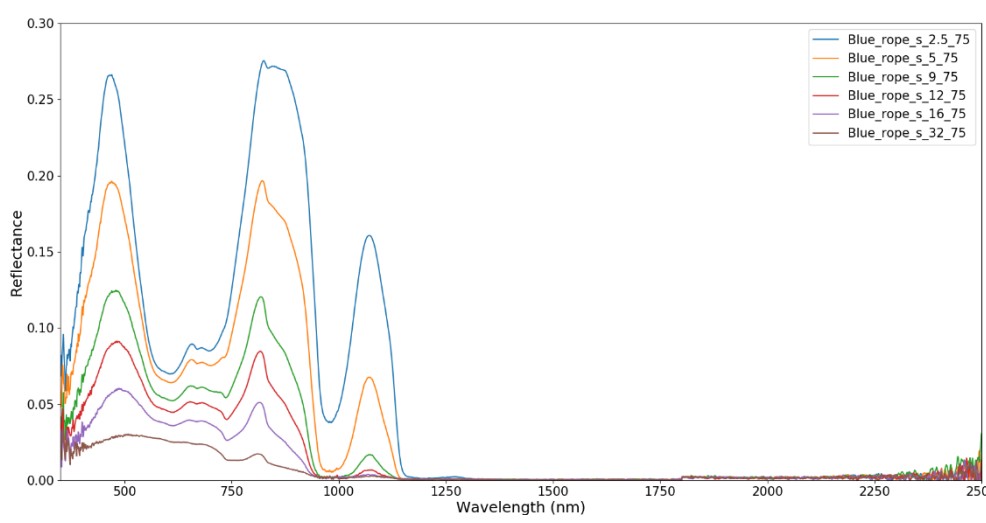

c)





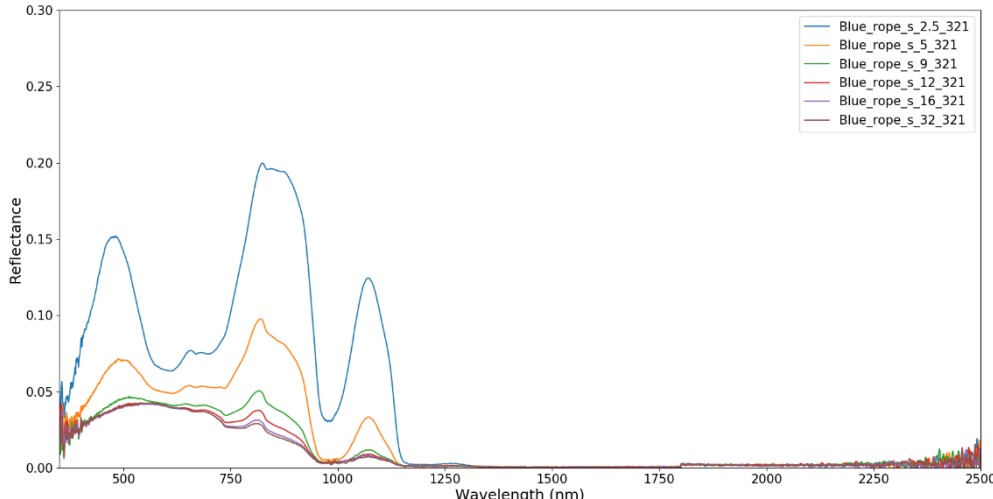

**Figure 9. Spectral reflectance of the blue rope submerged in a) clear water, b) water with a TSM concentration of 75 mg/L and c) water with a TSM concentration of 321 mg/L.**

### 4.2.4 Bags, cups and bottles

The spectral reflectance of plastic cups, bottles and bags was less uniform than the solid plastic samples shown earlier (ropes and placemats). Cups and bottles have different polymeric composition, HD-PE versus PET, which is also reflected in the absorption features in the SWIR (**Figure 10**). We observed variations in the magnitude and shape of reflectance of the cup, bag and bottle that were influenced by the condition of the items as wrinkled, crushed, straightened or containing a fluid (**Figure 11**).

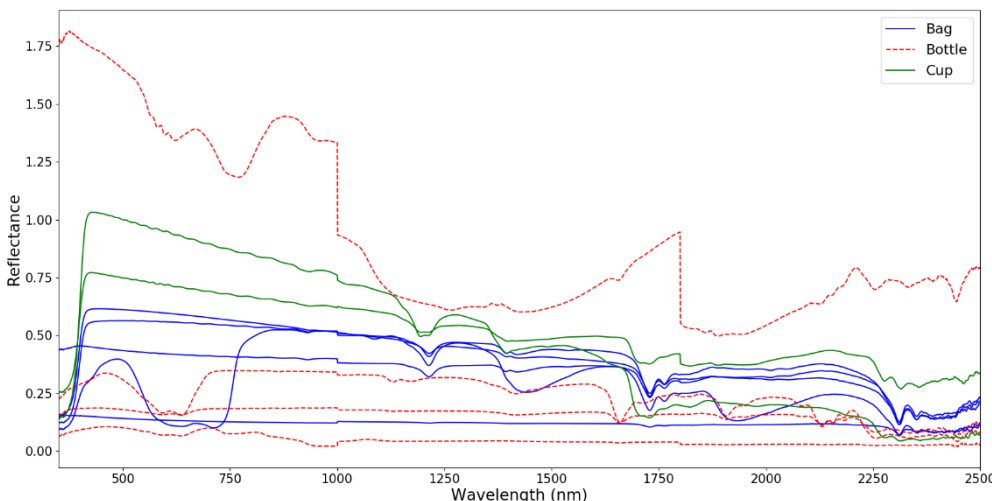



**Figure 10. Spectral reflectance of bags and bottles**

**Figure 11. Mean and standard deviation for the bags and bottles**

**4.3 Comparison ASD and SEV**

10    Figures 12 and 13 show the comparison of the spectral reflectance measurements made with the ASD and SEV spectroradiometers in the tank. Figure 12 shows the results for the blue rope aligned along the frame and Figure 13 shows the





results for the orange placemat. The plastic specimen were measured dry, wet and submerged at 2,5 cm, 5 cm, 9 cm, 12 cm, 16 cm and 32 cm without adding any sediments in the tank. The Unbiased Percentage Difference (UPD) was calculated for the dry specimen. The datasets agree very well with, for the dry measurements, the UPD mostly below 10%. At short ( < 400 nm) and long ( > 2240 nm) wavelengths larger differences can be observed due to the lower sensitivity of the instruments and

5 noisy spectra. Small differences can also be explained by a slightly different position of the two instruments. Although mounted closely, the footprint of the two instruments on the water surface will be slightly different.

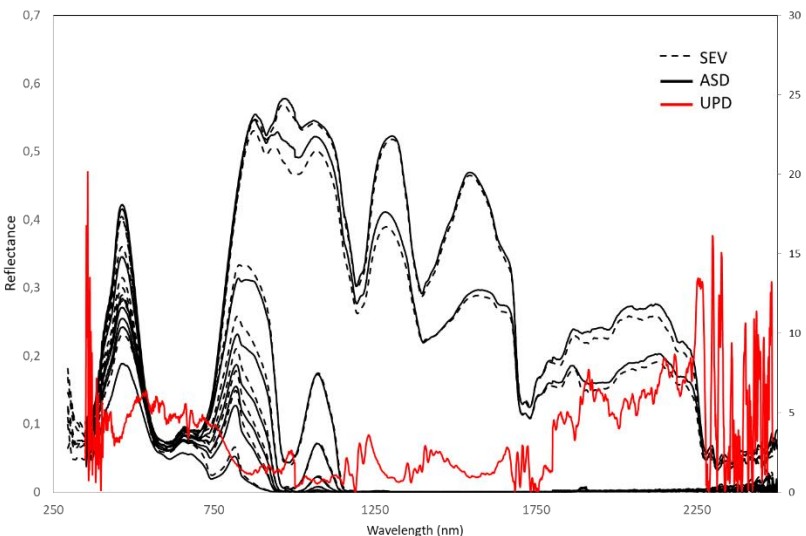

**Figure 12. Comparison of ASD and SEV with Unbiased Percentage Difference (UPD) for the blue rope.**



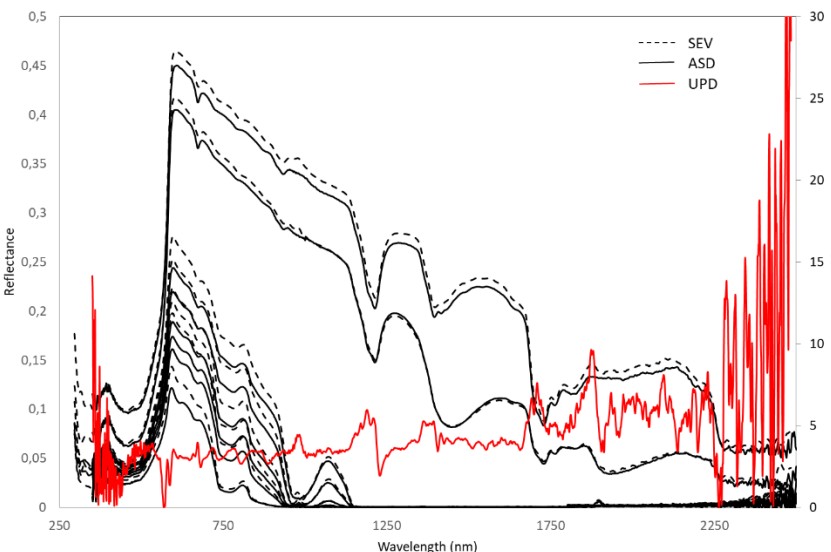

**Figure 13. Comparison of ASD and SEV with Unbiased Percentage Difference (UPD) for the orange placemat**

Figure 14 shows the UPD for the dry, wet and submerged orange placemat for wavelengths from 400 to 900 nm. The UPD is

5    higher for the submerged plastic samples and it generally increases with deeper submersion of the samples. Where the UPD

for the dry plastic is below 5% for this spectral range, the UPD for the plastic submerged at 16cm lies between 15 and 30%.

This can be explained by the footprint of the spectrometer which will increase and will be slightly different for the two

spectrometers as submersion depth increases.



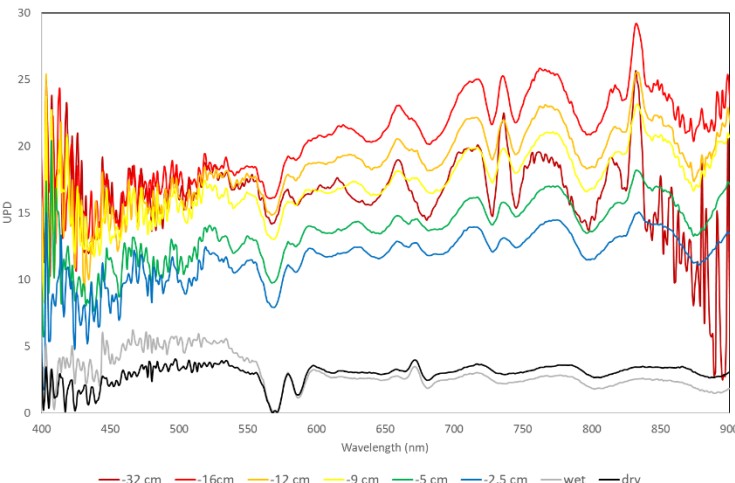

**Figure 14. Unbiased Percentage Difference (UPD) for the orange placemat, measured at different depths**

## 5 Data availability

All the datasets are available in a public data repository.

ASD measurements: https://doi.org/10.4121/12896312.v2 (Knaeps et al., 2020)

SEV measurements: https://doi.org/10.4121/uuid:9ee3be54-9132-415a-aaf2-c7fbf32d2199 (Garaba et al., 2020)

## 6 Conclusions

The results have shown that high quality spectral reflectance measurements were made in the VITO calibration facility and tank at Flanders Hydraulics. The dataset includes a large variety of plastic specimen, measured in dry and wet conditions, and

submerged in the tank. Submerging of plastics was done in a controlled way. Only plastics which were flat and homogeneous were submerged in the tank and their spectral reflectance was measured. Although these plastics are not found in the same way in nature, the results provide insights in the effect of water absorption and suspended sediments on the measured reflectance.  It shows the complexity of measuring plastics in a marine environment. Even more complexity is expected in an outdoor environment when surface features (glint, white caps) and changing light conditions come into play. The datasets

described here are made publicly available and the authors encourage scientists to use and further explore potential application of the datasets in developing remote sensing of marine litter relevant algorithms.



## Author contribution

EK and SS designed the experiments. GS designed the construction on top of the tank. EK, JM, SS, GS and SPG performed the spectral measurements in the calibration facility and the tank. MM (ASD) and SPG (SEV) prepared the spectral reflectance databases. DM performed the TSM measurements. EK prepared the manuscript with contributions from all co-authors.

## Competing interests

The authors declare that they have no conflict of interest.

## Financial support

This project has received funding from the ATTRACT project funded by the EC under Grant Agreement 777222. SPG was supported by the Deutsche Forschungsgemeinschaft (grant no. 417276871) and Discovery Element of the European Space
Agency's Basic Activities (ESA contract no. 4000132037/20/NL/GLC).

## Acknowledgements

We would like to thank the Port of Antwerp for providing the plastic litter samples and Robin de Vries for supporting the SEV spectral measurements.

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
