# Peer review of "Hyperspectral reflectance dataset of dry, wet and submerged marine litter"

_Earth System Science Data, 2020_

## Referee Comment (RC1) · Konstantinos Topouzelis (Referee) · 26 Nov 2020

This a very well-organized paper, presenting accurately and in details the field measurements and the collected dataset. Hyperspectral reflectance of marine litters is currently necessary to the scientific community for developing algorithms for detection and plastics discrimination. My only minor comment is about the 3.2. session on TSM measurements. I propose the authors to give some more information on the sentiment description (page4 line 12), e.g. on the small amount of clay and their decision to stop adding it into the tank.

---

## Short Comment (SC1) · 1 Dec 2020

Thank you for the comments on the paper. As already described, the cohesive sediment had a median particle size D50 of $11.0 \pm 0.3$ $\mu$m and ranged between a D10 of 2 $\mu$m to a D90 10 of 51 $\mu$m. The goal was to obtain different concentrations of suspended sediment to analyze the effects on the spectral reflectance of submerged plastics. Therefore we decided to add different amounts of the sediment in the water resulting in a TSM concentration of 75 mg L-1 and 321 mg L-1. We believe these two cases, next to the clear water case, provide sufficient data to analyze the effect of turbidity on the water reflectance

2020.

---

## Referee Comment (RC2) · Anonymous Referee #2 · 5 Dec 2020

Globally, the paper is well written, well structured and present the spectral measurement of 47 marine litter plastic items with a spectrometer (the Analytical Spectral Devices (ASD) FieldSpec 4). In addition, a few plastic items have also been measured with a hyperspectral radiometer (the Spectral Evolution SR-3501). Three minor comments are reported here below aiming at improving the general quality of this study before its publication. 1. Characterization of marine litter found in the port. To complement figure 2, it will be interesting if the authors could add a table showing the abundance and percentage of the marine litter items found in the port. 2. Page 2, line 5. Please specify which criteria was used to choose the plastic items that were measured with the hyperspectral radiometer. 3. Conclusions. I found that some conclusions fail to provide the key information that makes this dataset appellative for other

researchers to use it. For example, it would be an added value if the authors could specify which insights could be drawn from this study.

---

## Referee Comment (RC3) · Anonymous Referee #3 · 7 Dec 2020

Overall, the article is interesting, well written, clear and important to the scientific community given the scenario of debris pollution. I only have a minor suggestion and a comment. 1) Would be better to have the two datasets (Knaeps et al., 2020 and Garaba et al., 2020) in the same format (.txt or .xlsx) and to have a similar organization of the two datasets. I suggest harmonizing the two datasets. 2) Why you did not show and discuss the results of the "real" litter measurements in the results and discussion section?

---

## Author Comment (AC1) · 13 Jan 2021

We would like to thank all the referees for their comments. Our answers are provided below.

Konstantinos Topouzelis (Referee) topouzelis@marine.aegean.gr This a very well-organized paper, presenting accurately and in details the field measurements and the collected dataset. Hyperspectral reflectance of marine litters is currently necessary to the scientific community for developing algorithms for detection and plastics discrimination. My only minor comment is about the 3.2. session on TSM measurements. I propose the authors to give some more information on the sentiment description (page4 line 12), e.g. on the small amount of clay

[Figure]

and their decision to stop adding it into the tank.

Author Response

Thank you for the comments on the paper. As already described, the cohesive sediment had a median particle size D50 of $11.0 \pm 0.3$ $\mu$m and ranged between a D10 of 2 $\mu$m to a D90 10 of 51 $\mu$m. The goal was to obtain different concentrations of suspended sediment to analyse the effects on the spectral reflectance of submerged plastics. Therefore we decided to add different amounts of the sediment in the water resulting in a TSM concentration of 75 mg L-1 and 321 mg L-1. We believe these two cases, next to the clear water case, provide sufficient data to analyse the effect of turbidity on the water reflectance.

Anonymous Referee #2 Globally, the paper is well written, well structured and present the spectral measurement of 47 marine litter plastic items with a spectrometer (the Analytical Spectral Devices (ASD) FieldSpec 4). In addition, a few plastic items have also been measured with a hyperspectral radiometer (the Spectral Evolution SR-3501). Three minor comments are reported here below aiming at improving the general quality of this study before its publication. 1. Characterization of marine litter found in the port. To complement figure 2, it will be interesting if the authors could add a table showing the abundance and percentage of the marine litter items found in the port. 2. Page 2, line 5. Please specify which criteria was used to choose the plastic items that were measured with the hyperspectral radiometer. 3. Conclusions. I found that some conclusions fail to provide the key information that makes this dataset appellative for other researchers to use it. For example, it would be an added value if the authors could specify which insights could be drawn from this study.

Author Response

1. The abundance and percentage of plastic items in the Port of Antwerp are not known yet. Generally, it was observed by limited visual inspection, that the macrolitter

was mainly composed of plastic bags, ropes and driftwood. 2. There were several criteria: First of all, the goal was to collect plastics that are quite commonly found in rivers and coastal areas (plastic bags, ropes, bottles). Secondly, we wanted to include different colors to understand the effect on the spectral reflectance. Finally, the goal was to have several plastics that can be submerged in a controlled way in the tank (placemats and ropes that could be tied along the frame). 3. We want to point to an additional publication for Nature Scientific reports which is currently under review and which contains an extensive analysis of the dataset. The goal of this paper was to describe the dataset. However we will add some more insights on how the dataset can be used and why it is appellative to use it.

Text Added

Page 2, Line 5: Several criteria were used to select the plastic items. First of all, the plastics should be representative for the plastics that are commonly found in rivers and coastal areas (plastic bags, ropes, bottles). Secondly, plastics with different colors were included to understand the effect on the spectral reflectance (ropes and placemats in different colors). Finally, several plastics were added which could be submerged in a controlled way in the tank (placemats and ropes that could be tied along the frame).

Conclusion section: Remote Sensing datasets, either collected by satellites, drones or fixed cameras have a great potential to provide a better view on the marine plastics distribution and quantities. There is however a lack of knowledge on the optical properties of marine plastics in order to design appropriate detection algorithms. In particular for satellite data, where a satellite pixels has a footprint in the order of 5 to 20m, the observed spectral reflectance is often a mixture of plastics, water and surface features. It is very important to understand the spectral reflectance of marine plastics and how this spectral reflectance changes when the plastics are wet or slightly submerged. It is important to understand the differences with the reflectance of the water itself. For example, a turbid water plume can be easily misinterpreted for a brownish plastic sample. This datasets contributes to a better understanding of the optical properties of marine

plastic litter and encourages scientists to use the dataset in developing remote sensing of marine litter relevant algorithms.

Anonymous Referee #3 Overall, the article is interesting, well written, clear and important to the scientific community given the scenario of debris pollution. I only have a minor suggestion and a comment. 1) Would be better to have the two datasets (Knaeps et al., 2020 and Garaba et al., 2020) in the same format (.txt or .xlsx) and to have a similar organization of the two datasets. I suggest harmonizing the two datasets. 2) Why you did not show and discuss the results of the "real" litter measurements in the results and discussion section?

Author Response 1) We understand and agree with the Reviewer that it would have been better for the datasets to share the same format. Unfortunately, changing the current datasets would not be possible according to the repository guidelines. 'Be aware that once your dataset is published and is provided a DOI, the dataset cannot be altered or supplemented by additional data.' We have included a sentence in the conclusion section to urge future submission to adopt a uniform format and make sure metadata is provided.

2) The real litter measurements from the port are included in the Nature Scientific Reports paper.

Text Added The 4th Evolving and Sustaining Ocean Best Practices Workshop held in September 2020 the key outcome that was echoed in the session towards best practices for remote sensing of marine litter was open access to data and the need for unified processing methodologies and data presentation or formats. We recommend that experts in data collection and processing provide guidelines for sharing spectral and imagery relevant to marine litter or other ocean variables

In addition to the referee comments above, the following remarks and additions will be made to the manuscript:

- New reference added

New text Page 1 Line 24: To this end, there has been a rising interest in establishing spectral reference libraries of plastic litter in different conditions, states, types, pixel coverage and observation geometries (Garaba et al., 2020 In Press).

Reference Garaba, S. P., Arias, M., Corradi, P., Harmel, T., de Vries, R., and Lebreton, L. (2020 In Press) Concentration, anisotropic and apparent colour effects on optical reflectance properties of virgin and ocean-harvested plastics. Journal of Hazardous Materials, p. 124290, https://doi.org/10.1016/j.jhazmat.2020.124290

- We explain and showcase how the data can be further processed to correct for the jumps in the spectra using a splice correction. This splice correction is important especially when you compare the data for similarity using SAM or other statistical approaches.

New text Page 15 Line 2: In some of the spectra, a small jump can be observed around 1000 nm, and a second smaller one around 1750 nm. This is caused by the design of the ASD, consisting of three spectrometers and a fiber optic which is actually a bundle of 57 individual fibers, randomly oriented (Analytical Spectral Devices, 1999). VNIR and SWIR fibers are organized differently which makes that different areas of the surface are observed with different parts of the spectrum" resulting in steps in spectra at the joins between each detector (Arthur et al., 2012). This effect is observed when targets are non-uniform. It was decided not to apply any correction on the spectra in the dataset because the jumps can provide information on the uniformity of the target. However, readers can apply a splice correction themselves, which might be useful when readers apply further analysis on the data. A splice correction removes or compensates for the steps in spectra due to overlaps by the different detectors VNIR (350–1000 nm), SWIR-1 (1000–1800 nm), SWIR-2 (1800–2500 nm) The VNIR and SWIR-2 data can be adjusted to match the SWIR-1 data. The difference at 1000nm and 1001nm can be used to correct the VNIR data whilst the difference at 1800nm and

1801nm can be used to correct the SWIR-2 data

- In Figure 7, we will remove the spectrum with reflectance values above 1. All figures will be made uniform in terms of the axis and font size

---

## Author Response (AR1)

We would like to thank all the referees for their comments. Our answers are provided below.

Konstantinos Topouzelis (Referee) topouzelis@marine.aegean.gr
This a very well-organized paper, presenting accurately and in details the field measurements and the collected dataset. Hyperspectral reflectance of marine litters is currently necessary to the scientific community for developing algorithms for detection and plastics discrimination. My only minor comment is about the 3.2. session on TSM measurements. I propose the authors to give some more information on the sentiment description (page4 line 12), e.g. on the small amount of clay and their decision to stop adding it into the tank.

**Author Response**

Thank you for the comments on the paper. As already described, the cohesive sediment had a median particle size D50 of 11.0 ± 0.3 µm and ranged between a D10 of 2 µm to a D90 10 of 51 µm. The goal was to obtain different concentrations of suspended sediment to analyse the effects on the spectral reflectance of submerged plastics. Therefore we decided to add different amounts of the sediment in the water resulting in a TSM concentration of 75 mg L-1 and 321 mg L-1. We believe these two cases, next to the clear water case, provide sufficient data to analyse the effect of turbidity on the water reflectance.

Anonymous Referee #2
Globally, the paper is well written, well structured and present the spectral measurement of 47 marine litter plastic items with a spectrometer (the Analytical Spectral Devices (ASD) FieldSpec 4). In addition, a few plastic items have also been measured with a hyperspectral radiometer (the Spectral Evolution SR-3501). Three minor comments are reported here below aiming at improving the general quality of this study before its publication. 1. Characterization of marine litter found in the port. To complement figure 2, it will be interesting if the authors could add a table showing the abundance and percentage of the marine litter items found in the port. 2. Page 2, line 5. Please specify which criteria was used to choose the plastic items that were measured with the hyperspectral radiometer. 3. Conclusions. I found that some conclusions fail to provide the key information that makes this dataset appellative for other researchers to use it. For example, it would be an added value if the authors could specify which insights could be drawn from this study.

**Author Response**

1.  The abundance and percentage of plastic items in the Port of Antwerp are not known yet. Generally, it was observed by limited visual inspection, that the macrolitter was mainly composed of plastic bags, ropes and driftwood.
2.  There were several criteria: First of all, the goal was to collect plastics that are quite commonly found in rivers and coastal areas (plastic bags, ropes, bottles). Secondly, we wanted to include different colors to understand the effect on the spectral reflectance. Finally, the goal was to have several plastics that can be submerged in a controlled way in the tank (placemats and ropes that could be tied along the frame).
3.  We want to point to an additional publication for Nature Scientific reports which is currently under review and which contains an extensive analysis of the dataset. The goal of this paper was to describe the dataset. However we will add some more insights on how the dataset can be used and why it is appellative to use it.

**Text Added**

Page 4, Line 5:
Several criteria were used to select the plastic items. First of all, the plastics should be representative for the plastics that are commonly found in rivers and coastal areas. Secondly, plastics with different colors should be included to understand the effect on the spectral reflectance. Finally, several plastics should be selected which are uniform and can be submerged in a controlled way in the tank.
The above criteria resulted in a set of real/weathered and virgin plastics.

Conclusion section:
Remote Sensing images, either collected by satellites, drones or fixed cameras have a great potential to provide a better view on the marine plastics distribution and quantities. There is however a lack of knowledge on the optical properties of marine plastics in order to design appropriate detection algorithms and process these images into actionable maps for decision making. This spectral reflectance dataset contributes to a better understanding of the optical properties of marine plastic litter and encourages scientists to use the dataset in developing remote sensing of marine litter relevant algorithms.

The results have shown that high quality spectral reflectance measurements were made in the VITO calibration facility and tank at Flanders Hydraulics. The dataset includes a large variety of plastic specimen, measured in dry and wet conditions, and submerged in the tank. Submerging of plastics was done in a controlled way. Only plastics which were flat and homogeneous were submerged in the tank and their spectral reflectance was measured. Although these plastics are not found in the same way in nature, the results provide insights in the effect of water absorption and suspended sediments on the measured reflectance. It allows researchers to select the most appropriate wavelengths and prevent false detections. For instance, a turbid water plume can be easily misinterpreted for a brownish plastic sample. A deeper submersion of plastics might be misinterpreted as a decrease in the abundance of plastics.
The gathered dataset also shows the complexity of measuring plastics in a marine environment. Even more complexity is expected in an outdoor environment when surface features (glint, white caps) and changing light conditions come into play.

Anonymous Referee #3
Overall, the article is interesting, well written, clear and important to the scientific community given the scenario of debris pollution. I only have a minor suggestion and a comment. 1) Would be better to have the two datasets (Knaeps et al., 2020 and Garaba et al., 2020) in the same format (.txt or .xlsx) and to have a similar organization of the two datasets. I suggest harmonizing the two datasets. 2) Why you did not show and discuss the results of the "real" litter measurements in the results and discussion section?

**Author Response**
1)
We understand and agree with the Reviewer that it would have been better for the datasets to share the same format.
Unfortunately, changing the current datasets would not be possible according to the repository guidelines.
'Be aware that once your dataset is published and is provided a DOI, the dataset cannot be altered or supplemented by additional data.'
We have included a sentence in the conclusion section to urge future submission to adopt a uniform format and make sure metadata is provided.

2)
The real litter measurements from the port are included in the Nature Scientific Reports paper.

In addition to the referee comments above, the following remarks and additions will be made to the manuscript:

- New reference added

New text in red
**Page 1 Line 24:**
To this end, there has been a rising interest in establishing spectral reference libraries of plastic litter in different conditions, states, types, pixel coverage and observation geometries (Garaba et al., 2020 In Press).

**Reference**
Garaba, S. P., Arias, M., Corradi, P., Harmel, T., de Vries, R., and Lebreton, L. (2020 In Press) Concentration, anisotropic and apparent colour effects on optical reflectance properties of virgin and ocean-harvested plastics. Journal of Hazardous Materials, p. 124290, https://doi.org/10.1016/j.jhazmat.2020.124290

- We explain and showcase how the data can be further processed to correct for the jumps in the spectra using a splice correction. This splice correction is important especially when you compare the data for similarity using SAM or other statistical approaches.

New text in red
Page 15 Line 2:
In some of the spectra, a small jump can be observed around 1000 nm, and a second smaller one around 1750 nm. This is caused by the design of the ASD, consisting of three spectrometers and a fiber optic which is actually a bundle of 57 individual fibers, randomly oriented (Analytical Spectral Devices, 1999). VNIR and SWIR fibers are organized differently which makes that different areas of the surface are observed with different parts of the spectrum" resulting in steps in spectra at the joins between each detector (Arthur et al., 2012). This effect is observed when targets are non-uniform. It was decided not to apply any correction on the spectra in the dataset because the jumps can provide information on the uniformity of the target. However, readers can apply a splice correction themselves, which might be useful when readers apply further analysis on the data. The splice correction removes or compensates for the steps in the spectra. The VNIR and SWIR-2 data can be adjusted to match the SWIR-1 data. The difference at 1000 nm and 1001 nm can be used to correct the VNIR data whilst the difference at 1800 nm and 1801 nm can be used to correct the SWIR-2 data.